# Interspecific Hybridization and Complete Mitochondrial Genome Analysis of Two Ghost Moth Species

**DOI:** 10.3390/insects12111046

**Published:** 2021-11-21

**Authors:** Hua Wu, Li Cao, Meiyu He, Richou Han, Patrick De Clercq

**Affiliations:** 1Department of Plants and Crops, Faculty of Bioscience Engineering, Ghent University, Coupure Links 653, 9000 Ghent, Belgium; wuhuaflower@126.com; 2Guangdong Key Laboratory of Animal Conservation and Resource Utilization, Guangdong Public Laboratory of Wild Animal Conservation and Utilization, Institute of Zoology, Guangdong Academy of Science, Guangzhou 510260, China; caol@giabr.gd.cn (L.C.); hemeiyu_xw7@163.com (M.H.)

**Keywords:** *Thitarodes/Hepialus*, *Ophiocordyceps sinensis*, hybridization, infection, mitochondrial genome

## Abstract

**Simple Summary:**

The Chinese cordyceps is a valuable parasitic *Ophiocordyceps sinensis* fungus–*Thitarodes*/*Hepialus* larva complex. In view of culturing this complex, a method for the artificial rearing of the *Thitarodes*/*Hepialus* ghost moth hosts was established. Deterioration of the host insect population and low mummification rates in infected larvae constrain effective cultivation. Hybridization of *Thitarodes/Hepialus* populations may overcome this problem. *Thitarodes shambalaensis* and *Thitarodes* sp. were inbred or hybridized, and the biological parameters, larval sensitivity to the fungal infection and mitochondrial genomes of the resulting populations were investigated. Hybridization of *T. shambalaensis* and *Thitarodes* sp. allowed producing a new generation. One hybrid population (*T. shambalaensis* females mated with *Thitarodes* sp. males) showed increased population growth as compared with the parental *Thitarodes* sp. population. The sensitivity of the inbred larval populations to four fungal isolates of *O. sinensis* differed. The complete mitochondrial genomes of *T. shambalaensis*, *Thitarodes* sp. and the hybrid population were 15,612 bp, 15,389 bp and 15,496 bp in length, respectively. A + T-rich regions were variable in sizes and repetitive sequences. The hybrid population was located in the same clade with *T. shambalaensis*, implying the maternal inheritance of mitochondrial DNA.

**Abstract:**

The Chinese cordyceps, a parasitic *Ophiocordyceps sinensis* fungus–*Thitarodes/Hepialus* larva complex, is a valuable biological resource endemic to the Tibetan Plateau. Protection of the Plateau environment and huge market demand make it necessary to culture this complex in an artificial system. A method for the large-scale artificial rearing of the *Thitarodes/Hepialus* insect host has been established. However, the deterioration of the insect rearing population and low mummification of the infected larvae by the fungus constrain effective commercial cultivation. Hybridization of *Thitarodes/Hepialus* populations may be needed to overcome this problem. The species *T. shambalaensis* (GG♂ × GG♀) and an undescribed *Thitarodes* species (SD♂ × SD♀) were inbred or hybridized to evaluate the biological parameters, larval sensitivity to the fungal infection and mitochondrial genomes of the resulting populations. The two parental *Thitarodes* species exhibited significant differences in adult fresh weights and body lengths but not in pupal emergence rates. Hybridization of *T. shambalaensis* and *Thitarodes* sp. allowed producing a new generation. The SD♂ × GG♀ population showed a higher population trend index than the SD♂ × SD♀ population, implying increased population growth compared with the male parent. The sensitivity of the inbred larval populations to four fungal isolates of *O. sinensis* also differed. This provides possibilities to create *Thitarodes*/*Hepialus* populations with increased growth potential for the improved artificial production of the insect hosts. The mitochondrial genomes of GG♂ × GG♀, SD♂ × SD♀ and SD♂ × GG♀ were 15,612 bp, 15,389 bp and 15,496 bp in length, with an A + T content of 80.92%, 82.35% and 80.87%, respectively. The A + T-rich region contains 787 bp with two 114 bp repetitive sequences, 554 bp without repetitive sequences and 673 bp without repetitive sequences in GG♂ × GG♀, SD♂ × SD♀ and SD♂ × GG♀, respectively. The hybrid population (SD♂ × GG♀) was located in the same clade with GG♂ × GG♀, based on the phylogenetic tree constructed by 13 PCGs, implying the maternal inheritance of mitochondrial DNA.

## 1. Introduction

The Chinese cordyceps (*Ophiocordyceps*
*sinensis* fungus–*Thitarodes* larva complex) is a valued biological resource endemic to the Tibetan Plateau and widely used in medicinal treatments including fatigue, asthma, respiratory and kidney diseases and as health foods, especially in many Asian countries [1,2,3,4]. The annual yield of the natural Chinese cordyceps has decreased sharply in recent years due to over-exploitation and habitat degradation [5,6]. Due to the extremely limited wild resource and huge market demand, this wild fungus–insect complex costs USD 60,000–75,000 per kilogram [7,8,9,10,11]. Artificial cultivation of the Chinese cordyceps is necessary to protect this valuable bio-resource and to supply commercial trade [4,11].

The insect host species of *O. sinensis* belong to the family Hepialidae (Lepidoptera). *Thitarodes armoricanus* Oberthür was the first host insect of *O. sinensis* in China to be reported and biologically characterized [12]. Research on the diversity and taxonomy of Hepialidae expanded rapidly during the 1980s. Abundant diversity and phylogeographical structures for the host insects of *O*. *sinensis* have been reported [13,14]. Comparative phylogenetic analyses have suggested coevolutionary relationships between *O. sinensis* and its host insects [14,15]. It appears that most host insect species of the *O. sinensis* fungus have a very narrow distribution on the Tibetan Plateau, and host insect species might vary among different mountain ranges and even from different sides and habitats of the same mountain [16]. Different insect host species or strains show significant differences in morphology, biology and ecology [16,17,18,19,20]. Long-distance dispersal of host insects seems very limited due to the short lifespan of the adult and the heavy abdomens of females that carry eggs; thus most shared genotypes are between strains from adjacent geographic regions [14]. Given the complex and harsh ecological environment in the Tibetan Plateau, most Hepialidae species have a narrow-area distribution type [21]. Of ninety-one named Hepialidae insects spanning thirteen genera reported to be related to host insects of the *O. sinensis* fungus, fifty-seven are considered potential host species of the fungus and are distributed throughout the Tibetan Plateau [17]. However, the described insect host species or strains of *O. sinensis* have never been confirmed by a hybridization technique. 

Artificial cultivation of the Chinese cordyceps in low-altitude areas is successful and contains three important milestones: mass production of the host insects, cultivation of effective *O. sinensis* fungus and formation of a fruiting body from the infected insect cadaver. The method for the large-scale artificial rearing of the *Thitarodes*/*Hepialus* insect hosts is established [11,19,20]. Culture parameters such as food, temperature, humidity and culturing time influence the efficiency of artificial cultivation of different insect species and stages [19,20]. The natural foods of *Thitarodes*/*Hepialus* larvae are tender roots and buds of plants of the Polygonaceae family such as *Polygonum viviparum*, *P. sphaerostachyum* and *P. capitatum*, as well as other plants in nine different families (e.g., Ranunculaceae, Juncaceae and Cyperaceae) [22]. The roots of *Potentilla anserina* and carrots (*Daucus carota*) are favorite larval foods for indoor cultivation [4,23]. *Thitarodes* species have long and unusual life cycles; it takes 263 to 494 days for *Thitarodes jianchuanensis* and 443 to 780 days for *T. armoricanus* to complete the life cycle, including egg, larval instars L1-L9, pupa and adult; the larvae can develop into pupae from the L7, L8 or L9 instar [20]. During the long life cycle and several successive generations in the culture rooms, the insects are usually prone to deterioration [19,24]. How to overcome this common phenomenon during insect host rearing is a key issue for the commercial production of the Chinese cordyceps. 

Various *O. sinensis* strains from different locations in the Tibetan Plateau have been isolated [2,14,25] and cultured in solid media and liquid media to obtain conidia and blastospores [26]. Stable fruiting body production of *O. sinensis* with mature ascospores by artificial media without living insects has also been successfully realized [25]. Two efficient methods are used to infect the host larvae with the *O. sinensis* fungus: the larvae are routinely infected by the mature ascospores collected from the wild Chinese cordyceps or from the artificial fruiting bodies, and the larvae are injected into the hemocoel with the blastospores from the liquid culture containing maltose as a carbon source [26]. In the hemolymph of the larvae, the spindle blastospores exhibit a dimorphic developmental process [26] and may produce exponentially by budding growth and/or grow into elongate hyphal bodies (pseudohyphae) and hyphae by apical growth under the induction of unknown factors [26,27], as reported in the dimorphic fungi *Candida albicans* and *Ustilago maydis* [28] and the entomopathogenic fungus *Metarhizium rileyi* [29]. The living infected host larvae might harbor the spindle blastospores in the hemolymph for several months, contrary to other entomopathogenic fungi such as *M. anisopliae* and *Beauveria bassiana* which cause the death of their host larvae within a few days [30,31]. The slow mummification of the larvae post infection is an obstacle for cost-efficient production of Chinese cordyceps [4,8,11]. Selection of an *O. sinensis* fungus isolate with high mummifying potential and a *Thitarodes* insect host species or strain with high sensitivity to the fungal infection should contribute to overcoming this obstacle in the artificial production of the Chinese cordyceps.

Mitochondrial genome sequences have been widely used as molecular markers for diverse evolutionary analyses because of their unique features, including coding content conservation, maternal inheritance and rapid evolution [32]. Insect mitochondrial genomes are usually small closed-circular molecules (15–20 kb) containing 13 protein-coding genes (PCGs), 2 ribosomal RNA (rRNA) genes, 22 transfer RNA (tRNA) genes, and a large non-coding element termed the A + T-rich or control region [21,33,34]. The mitochondrial gene order also provides important evidence for establishing genome evolutionary relationships [34,35]. Due to the improved sequencing technology, insect mitochondrial genomes have been heavily sequenced in recent years. Eight *Thitarodes*/*Hepialus* mitochondrial genomes are reported, including *Thitarodes renzhiensis* (accession number HM744694; size 16,173 bp), *Thitarodes yunnanensis* (former *Ahamus yunnanensis*) (accession number HM744695; size 15,816 bp) [21,36], *Thitarodes*
*pui* (accession numbers KF908880 and MK599283; sizes 15,064 bp and 15,928 bp) [21,37], *Hepialus*
*xiaojinensis* (accession number KT834973; size 15,397 bp) [38], *Hepialus gonggaensis* (accession number KP718817; size 15,940 bp) [39], *Thitarodes sejilaensis* (accession number KU053201; size 15,290 bp;) [40], *Thitarodes* sp. (accession number KX527574; size 16,280 bp) [41] and *Thitarodes damxungensis* (accession number MK648145; size 15,362 bp) [21]. With respect to a total of 57 recognizable potential host species of the fungus, the information of the mitochondrial genomes of existing ghost moths is still very limited, and no reports are available on the mitochondrial genomes from the hybrids. 

Insight into the biological and molecular characters of the inbred and hybrid populations is elementary for the effective artificial cultivation and evolutionary analysis of these *Thitarodes* insects. In this study, the hybridization between *T. shambalaensis* and an undescribed *Thitarodes* species from two different locations in the Tibetan Plateau was demonstrated. The fitness parameters (such as the number of eggs per female, egg hatching rates, larval fresh weights, larval survival rates, female and male pupal ratios, population trend indexes), larval sensitivity to the fungal infection and mitochondrial genomes of the resulting inbred and hybrid populations were determined to evaluate the hybridization effects. 

## 2. Materials and Methods 

### 2.1. Morphological and Molecular Characteristics of Thitarodes Insect Populations

The pupae of two *Thitarodes* insect populations were, respectively, from the mountains in Gongga (referred to as GG♂ × GG♀) (2476 m, 29°70′ N, 102°03′ E) and Shade (referred to as SD♂ × SD♀) (4560 m, 29°65′ N, 101°31′ E), Kangding in Sichuan Province, China. 

The valve pattern of the male genitalia is an important characteristic for the morphological identification of Hepialidae insects [42,43]. The female and male *Thitarodes* pupae were differentiated by their genitalia. Briefly, in the last abdominal segment, females exhibit a long longitudinal suture linked to the previous abdominal segment without papillary structures, whereas males exhibit a short longitudinal suture between two papillary structures that is not linked to the previous abdominal segment [44]. The males of GG♂ × GG♀ and SD♂ × SD♀ populations were dissected to show the valve patterns in the laboratory. For the molecular identification of these *Thitarodes* populations, Cytochrome b and *cox1* sequences were amplified with the primers CB1 (TATGTACTACCATGAGGACAAATATC) and CB2 (ATTACACCTCCTAATTTATTAGGAAT) [42,45] and LCO1490 (GGTCAACAAATCATAAAGATATTGG) and HCO2198 (TAAACTTCAGGGTGACCAAAAAATCA) [46], respectively.

### 2.2. Inbred and Hybrid Thitarodes Populations

Four inbred and hybrid combinations (GG♂ × GG♀, SD♂ × SD♀, SD♂ × GG♀, GG♂ × SD♀) were created with 50 female and 75 male adults for each combination, but the population GG♂ × SD♀ could not be established due to technical issues related to climatization of the culture room. Three replicates were set up for each combination. The male and female pupae were housed in cartons (L = 104 cm; W = 50 cm; H = 50 cm) with moist moss at 9–17 °C and 50–80% relative humidity. When the adults emerged, they were housed in small cylindric nets (D = 28 cm; H = 32 cm) to allow mating for 3–5 days. The collected eggs from the mated females were transferred to a culture room and maintained at 9–13 °C to establish the experimental populations in the Institute of Zoology, Dongguan (43 m above sea level), Guangdong Province, China.

To evaluate the development, survival, fertility and sensitivity to the fungal infection in the resulting *Thitarodes* populations, 600 eggs from each inbred or hybrid combination were surface-sterilized for 3 min with a solution containing 2.5 mL of 4 M NaOH, 0.5 mL of 12% NaOCl and 21.5 mL of distilled water [47], rinsed 3 times with sterile distilled water and placed in a sterile plastic container (L = 48 cm; W = 35 cm; H = 17 cm) containing 2 kg coconut peat (65% of water content) and 1 kg *Potentilla anserina* roots as food at 9–13 °C. For each hybridization combination, 30 containers were established. When the larvae reached the third instar, they were individualized into a plastic cup (D = 3.5 cm; H = 6.5 cm) with the same peat and food as above (15 g coconut peat and 15 g food for each cup) to avoid larval cannibalism [20]. Fresh food was added to each cup every 2 months to obtain 6th instar larvae (average fresh weight = 0.52 ± 0.03 g) for fungal infection by the injection method. The larval number in each container was recorded, and the average hatch rate was calculated. At the sample date (every 30 days), the survival rates, longevity, fresh weight, body length and sex proportion of pupae and adults and fecundity were recorded.

### 2.3. O. sinensis Fungal Isolates

KD, YN, XZ and QH fungal isolates of *O. sinensis* isolated from the fruiting bodies of wild Chinese cordyceps, respectively, from Sichuan, Yunnan, Tibet and Qinghai, China, were cultured on PPDA medium (liquid PPDA medium: 200 g potato extract, 20 g glucose, 10 g peptone, 1.5 g KH_2_PO_4_, 0.5 g MgSO_4_, 20 mg vitamin B_1_ and 1000 mL distilled water; solid PPDA medium: 15% agar in liquid PPDA medium) at 13 °C. The fungal isolates were identified by using the amplified sequence from the internal transcribed spacer (ITS; ITS1-5.8S-ITS2) of the nuclear ribosomal DNA as described by [48]. The identified *O. sinensis* isolates were preserved at −80 °C in the Institute of Zoology, Guangdong Academy of Science, Guangzhou, China.

The fungal colonies cultured on the PPDA plates at 13 °C for 60 days were transferred to 250 mL flasks containing 150 mL liquid PM medium (200 g potato extract, 20 g maltose, 10 g peptone, 1.5 g KH_2_PO_4_, 0.5 g MgSO_4_, 20 mg vitamin B_1_ and 1000 mL distilled water) [26]. The flasks were incubated on a 120 rpm shaker at 13 °C, the blastospores from the flasks were harvested after 50 days by using three layers of sterile lens papers to remove hyphae and large particles, and the filtered solution was centrifuged at 8000 rpm for 15 min at 10 °C. The harvested blastospores were re-suspended in sterile phosphate-buffered saline (PBS; pH 7.0) at a concentration of 3.0 × 10^6^ blastospores per mL and kept at 4 °C for less than 3 days before use for larval infection.

### 2.4. Larval Infection of Inbred Populations by O. sinensis Isolates

The larvae from 2 inbred populations (GG♂ × GG♀, SD♂ × SD♀) were injected with KD, YN, XZ or QH fungal isolates of *O. sinensis*. An aliquot of 4 µL blastospore suspension containing 1.2 × 10^4^ blastospores was injected into each 6th instar larva by a microinjection system (IM-31; Narishige, Tokyo, Japan). One hundred and eighty larvae were used for each replicate, and three replicates were set for each injection. Larvae injected with PBS buffer or without any injection were set as controls. The injected larvae were reared at 4 °C for one week and then transferred to a culture room at 13 °C. After 90 days, about 10 µL of hemolymph of each injected larva (6th instar) was sampled to confirm the presence of the growing blastospores stained by Calcofluor White (Sigma, Kanagawa, Japan) and observed by a fluorescence microscope (IX73; Olympus, Tokyo, Japan). The injected larvae were reared at 13 °C until the larvae became stiff and were coated with growing mycelia. The mummified larvae with head upward were then planted into soil of 55–60% humidity to induce the formation of stroma at 4 °C. The survival and mummification of the injected larvae were monthly checked. Data on larval infection of the hybrid populations could not be gathered due to an insufficient number of larvae available for fungal injection.

### 2.5. Analysis of the Mitochondrial Genomes

Three male adults from GG♂ and SD♂ and three larvae from SD♂ × GG♀ in dry ice were used for mitochondrial genome sequencing by Shanghai BIOZERON Co., Ltd., with the routine method [41]. The nucleotide sequences of protein-coding genes (PCGs) from the annotated mitochondrial genomes were translated to protein sequences using the invertebrate mitochondrial code. For the base composition of the nucleotide sequences, the composition skewness was calculated as follows: AT skew = [A − T]/[A + T], GC skew = [G − C]/[G + C] [49]. Thirteen PCGs and two rRNA genes were inferred based on comparison with mitochondrial genomes of 10 previously sequenced Hepialidae species (*T. damxungensis, T. gonggaensis, T. pui, T. renzhiensis, T. sejilaensis, Thitarodes* sp., *H. xiaojinensis, T. yunnanensis, Napialus hunanensis, Endoclita signifer*). The location and secondary structures of the 22 tRNA were predicted by tRNAscan-SE (http://lowelab.ucsc.edu/tRNAscan-SE/) (accessed on 13 November 2020). After the removal of the termination codon, the codon usage frequency and the first, second and third base use frequency of the codon were calculated using MEGA 7.0. The overlapping regions and intergenic spacers between genes were manually counted. The entire A + T-rich region was subjected to a search for the tandem repeats using the Tandem Repeats Finder program [50].

To construct the phylogenetic relationships within Hepialidae in Lepidoptera, 10 complete mitochondrial genomes of the above hepialid species were downloaded from GenBank. *Drosophila melanogaster* was used as an outgroup. A maximum likelihood (ML) tree was built in MEGA 7.0 using the nucleotide sequence of 13 PCGs, based on the “find best DNA/protein models (ML)”. The “GTR + G” model was chosen for phylogenetic analysis because it produced the lowest values for both the BIC (Bayesian information criterion) and the AICc (corrected Akaike information criterion). The confidence values of the ML tree were evaluated via a bootstrap test with 1000 iterations.

### 2.6. Data Analysis

The data are expressed as means ± SE. The average survival rates, fresh weight, body length and sex proportions of pupae and adults at the sample time points were determined. In addition, a population trend index (I) was calculated with I = P_II_/P_I_, where P_I_ = numbers of pupae in the parental generation, and P_II_ = numbers of pupae in the next generation [20]. The data were analyzed with SPSS 21.0 (SPSS Inc., Chicago, IL, USA) to compare the differences among the treatments. Differences among means by Tukey’s multiple-range test were considered significant at *p* < 0.05.

## 3. Results

### 3.1. Morphological and Molecular Identification of Two Thitarodes Species

The SD♂ × SD♀ population from Sichuan Province, China, was considered to be an undescribed *Thitarodes* species, by the phylogenetic analysis, although the valve pattern of the male genitalia of SD♂ × SD♀ resembled that of *Thitarodes kangdingensis* (Figure 1) (Prof. Zhiwen Zou, personal communication). The GG♂ × GG♀ population also from Sichuan Province, China, was confirmed to be *Thitarodes shambalaensis* [43], based on the valve pattern of the male genitalia (Figure 1) and the sequences of the *cox1* fragment. The complete *cox1* sequences of the two species were 1531 bp and were submitted to GenBank (accession numbers OK104111 and OK047724, respectively). 

### 3.2. Development from the Pupae to Next-Generation Pupae in Inbred and Hybrid Populations

The data on the fresh weight and body length of the pupae from the parental species were collected. The fresh weights were 0.88 ± 0.04 g and 0.67 ± 0.03 g for SD♂ × SD♀ and GG♂ × GG♀ female pupae and 0.59 ± 0.05 g and 0.45 ± 0.03 g for SD♂ × SD♀ and GG♂ × GG♀ male pupae, respectively. The body lengths were 2.73 ± 0.21 cm and 2.50 ± 0.05 cm for SD♂ × SD♀ and GG♂ × GG♀ female pupae and 2.46 ± 0.08 cm and 2.25 ± 0.04 cm for SD♂ × SD♀ and GG♂ × GG♀ male pupae, respectively (Table 1). The pupae developed into adults in 30–45 days at 9–17 °C. No significant differences were found for the fresh weights and body lengths of the pupae between the two populations, except for the fresh weight of female pupae, which differed significantly between the populations (Table 1). The ratios of females and males in the pupae (SD♂ × SD♀: 1.27 ± 0.16; GG♂ × GG♀: 1.04 ± 0.06) and adults (SD♂ × SD♀: 0.89 ± 0.06; GG♂ × GG♀: 0.87 ± 0.03) were also not significant. The emergence rates were 38.38% and 48.96% for SD♂ × SD♀ and GG♂ × GG♀ females and 43.57% and 57.58% for SD♂ × SD♀ and GG♂ × GG♀ males, respectively, showing no significant differences between the two parental insect populations (Table 1). 

The females and males of the GG♂ × GG♀ population were mating all day and night, like those of SD♂ × GG♀. However, those of SD♂ × SD♀ usually mated in the evening and at night. The adults did not feed, and their life span usually lasted 5–7 days at 9–17 °C. Contrary to the clean eggs from GG♂ × GG♀, the eggs from SD♂ × SD♀ were coated with a sticky secretion. The average number of eggs per female for SD♂ × SD♀ (512 ± 3) was significantly higher than that for the other populations, whereas the number of eggs in one milliliter (3976 ± 109) was higher and the weight of one thousand eggs (0.18 ± 0.01 g) was lower for SD♂ × SD♀ compared with other populations, indicating the smaller egg size of SD♂ × SD♀ (Appendix A). The hatching rate for SD♂ × GG♀ (12.62 ± 2.80%) was significantly lower than that for the other populations (Appendix A). Thus, these two insect populations exhibited characteristic differences in mating behavior, egg size and the presence of a sticky secretion coating the eggs but showed no significant differences in the ratio of female and male pupae or in the pupal emergence rates.

The GG♂ × GG♀ and SD♂ × SD♀ populations were hybridized in the culture room, and the larvae successfully became pupae in the resulting hybrid populations. The fresh weights of the larvae from the inbred and hybrid populations did not vary significantly with culture times in 12 months (Appendix A). The survival rates of the larvae were stable at 80–100% in the first 10 months but decreased sharply after 11 months, especially in the SD♂ × SD♀ and SD♂ × GG♀ populations (Appendix A), due to the increasing larval mortality before pupation. The larvae became pupae after 22 months. The ratios of the resulting female and male pupae were 0.28 ± 0.15 for SD♂ × SD♀, 0.61 ± 0.05 for SD♂ × GG♀ and 0.78 ± 0.03 for GG♂ × GG♀, which were significantly different. Population trend index values were quite variable, with I = 0.01 for SD♂ × SD♀, 0.32 for SD♂ × GG♀ and 2.25 for GG♂ × GG♀, indicating different proportions of pupal numbers in the previous generation over the next generation in the inbred and hybrid populations. The adults emerged from the pupae after 23 months. Unfortunately, the experiments were discontinued due to the emergence of too few adults caused by the high mortality (>70%) of the pupae. Nonetheless, the above results indicate that hybridization of *Thitarodes* sp. and *T. shambalaensis* allowed harvesting a next generation of adults at least from the SD♂ × GG♀ population.

### 3.3. Larval Infection of Inbred Populations by O. sinensis Isolates 

As shown in Appendix A, the percentages of the larvae carrying the blastospores varied at 120 days post infection, from 20.37 ± 5.38% for the larvae of GG♂ × GG♀ injected with fungal isolate XZ to 62.96 ± 1.96% for the larvae of GG♂ × GG♀ injected with fungal isolate QH; for SD♂ × SD♀, the mummification rate of the larvae containing fungal isolate XZ was significantly lower than that of those containing fungal isolates KD, QH and YN; for GG♂ × GG♀, the mummification rate of the larvae containing fungal isolate XZ was significantly lower than that of those containing fungal isolates KD, QH and YN. For fungal isolates KD, XZ and YN, no significant differences in mummification rate were observed among the two infected larval populations (Appendix A). Sixty and 90 days after infection, no significant differences in the percentages of the larvae carrying blastospores were found among both larval populations. Although the larval hemocoel was filled with growing blastospores after 90 days, the fresh weights of the larvae in both populations did not differ markedly (Appendix A).

### 3.4. Mitochondrial Genome Analysis

Organization and base composition. The complete mitochondrial genomes of two inbred and one hybrid *Thitarodes* populations (SD♂ × SD♀, GG♂ × GG♀ and SD♂ × GG♀) were a circular DNA molecule of 15,389 bp, 15,612 bp and 15,496 bp in length, respectively (accession number: MZ675586, MZ675587 and MZ675588) (Figure 2). Like most other metazoan mitochondrial genomes, each of three *Thitarodes* mitochondrial genomes contained 13 PCGs, 22 tRNAs, 2 rRNAs and a large non-coding control region. Among the 37 genes in each mitogenome, there were 9 PCGs and 14 tRNAs encoded in the heavy strand, while 4 PCGs, 8 tRNAs and 2 rRNAs were encoded in the light strand. The mitochondrial genome structure was compact. The gene order of the *Thitarodes* mitochondrial genomes was uniform (Figure 2).

The mitochondrial genome content of these three *Thitarodes* populations was A + T-biased, ranging from 80.87% (SD♂ × GG♀) to 82.35% (SD♂ × SD♀) (Table 2). The A + T content of the SD♂ × SD♀ mitochondrial genome was 82.35%, which was larger than that of the other two mitochondrial genomes. The AT skew in the forward strand of the SD♂ × SD♀ mitochondrial genome was slightly positive (0.008), which was different from the other two mitochondrial genomes (0.020) (Table 2). Likewise, The GC skew of the SD♂ × SD♀ mitogenome (−0.182) was also obviously different from those of the other two (−0.231 to −0.234).

Protein-coding genes. The 13 PCGs in these mitochondrial genomes included 7 *NADH* dehydrogenase subunits (*nad1-6*, *nad4L*), 3 cytochrome c oxidase subunits (*cox1-3*), 2 ATPase subunits (*atp6, atp8*) and one cytochrome b gene (*cytb*). The lengths of the 13 PCGs in the mitochondrial genomes of SD♂ × SD♀, GG♂ × GG♀ and SD♂ × GG♀ were 11,073, 11,067 and 11,067, respectively (Table 2). When the termination codons were excluded, the 13 PCGs in SD♂ × SD♀, GG♂ × GG♀ and SD♂ × GG♀ were composed of 3680, 3678 and 3678 codons, respectively. These findings indicate a high degree of similarity in the PCG code number among the three mitochondrial genomes (Figure 3). 

The codon frequency analysis of the SD♂ × SD♀, GG♂ × GG♀ and SD♂ × GG♀ mitochondrial genomes showed that a total of 61 codons were used for transcription, with the absence of UAG (Figure 3). The GG♂ × GG♀ mitochondrial genomes had 10 more codons than SD♂ × SD♀. However, the most frequently used codon in the three mitochondrial genomes was UUA for Leu, followed by AUU for Ile (Figure 3). 

The fraction of codons encoding the hydrophobic amino acids (Met, Trp, Phe, Val, Leu, Ile, Pro, Ala) in the mitochondrial genomes of SD♂ × SD♀, GG♂ × GG♀ and SD♂ × GG♀ were 56.91%, 56.60% and 56.62%, respectively (Figure 3), reflecting the biased usage of A/T nucleotides and the hydrophobic nature of respiratory membrane complexes. The codon distribution patterns of the three compared mitochondrial genomes (Figure 3) were consistent with the finding that Ile, Leu, Phe, Ser, Asn and Tyr are the six amino acids most frequently used, whereas Lys is rare in Hepialidae [51].

All PCGs (except *cox**1*) in the three mitochondrial genomes began with a canonical start codon (ATN or NTG). More specifically, eight PCGs (*cox3, atp6, atp8, nad1,*
*nad2, nad4L**, nad5* and *nad**6*) started with ATA, one PCG (*nad3*) with ATT, three PCGs (*nad4**, cytb* and *cox2*) with ATG and one PCG (*cox1*) with CGA (Table 2). For the stop codon, nine PCGs (*atp6, atp8, nad1, nad2, nad4, nad4L, nad6, cox3* and *cob*) were terminated with the typical stop codon TAA, while three PCGs *(nad**5,*
*cox1* and *cox2*) located upstream of tRNAs ended with T, and one PCG (*nad3*) ended with TAG (Table 2). 

Transfer RNA and ribosomal RNA genes. The predicted cloverleaf structures for 22 tRNA genes are presented in Appendix A. For three mitochondrial genomes of SD♂ × SD♀, GG♂ × GG♀ and SD♂ × GG♀, the *rrnL* (16S rRNA) and *rrnS* (12S rRNA) genes were identified, being 1355 bp, 1336 bp and 1336 bp and 777 bp, 778 bp and 778 bp in size, respectively, falling into the reported range for the Hepialidae (1324–1375 bp, 740–781 bp) [41] (Table 2). The *rrnL* gene was located between *trnL1* (TAG) and *trnV* (TAC), while *rrnS* was located between *trnV* (TAC) and the A + T-rich region (Table 2 and Figure 2). The A + T percentages of rRNA in three mitochondrial genomes were 85.10% to 85.60%. These rRNA characteristics are consistent with those observed in other Lepidoptera [41]. Twenty-two tRNAs were encoded in two mitochondrial genomes of the GG♂ × GG♀ and SD♂ × GG♀ populations, ranging from 60 bp to 73 bp in size and spread across the entire genome. SD♂ × SD♀ ranged from 61 bp to 71 bp. All tRNAs were shown to be folded into the expected clover-leaf secondary structure except for *trnS1* (UCU), which lacks the dihydrouridine (DHU) loop (Appendix A). This feature is common to most of the available lepidopteran mitochondrial genomes [52].

Non-coding and overlapping genes. The complete mitochondrial genomes of SD♂ × SD♀, GG♂ × GG♀ and SD♂ × GG♀ were very compact with a total of 198, 190 and 188 non-coding bp dispersed among 20, 17 and 17 pairs of neighboring genes ranging from 1 to 42 bp, 1 to 45 bp and 1 to 45 bp, respectively (Table 2). 

The longest spacer sequence was located between *nad5* and *trnH*. A 15 bp intergenic spacer located between the *trnS2* and *nad1* contained the “ATACTAA” motif, which is a common feature across lepidopteran insects [53,54], but in Hepialidae species, the non-coding region contained an “ATACTA” sequence followed by T or C (Figure 4). The results are consistent with the report from [40]. In addition, the complete mitochondrial genomes of SD♂ × SD♀, GG♂ × GG♀ and SD♂ × GG♀ were 42, 24 and 24 bp overlapping nucleotides located in 7, 10 and 10 pairs of neighboring genes with a length from 1 to 25 bp. The longest overlapping nucleotides (25 bp) existed between *trnL1* and *rrnL*. The *atp8* and *atp6* had 4 bp overlapping nucleotides (Appendix A). Those seven nucleotides “ATGATAA” are a common feature across the lepidopteran mitochondrial genomes [50].

A + T-rich region. The length and A + T content of the A + T-rich regions were 554 bp and 91.70% in SD♂ × SD♀, 787 bp and 89.83% in GG♂ × GG♀ and 673 bp and 90.64% in SD♂ × GG♀ (Table 2). The A + T-rich region was located between the *rrnS* and *trnI* genes. These repeat sequences accounted for some of the variations in mt genome length (Table 2). The A + T-rich region of *T. renzhiensis* is the longest of all the sequenced Lepidoptera mitochondrial genomes; the shortest is 319 bp in *O. lunifer* [55]. This variation in length could be related to the number and lengths of tandem repeats in the control region [56]. SD♂ × SD♀ had no repeating sequences (Table 2). Two repeated sequences (114 bp) in GG♂ × GG♀ were detected, each with one more base A. Compared with GG♂ × GG♀, SD♂ × GG♀ lacked a set of repeated sequences, and two repeated sequences were 113 bp and 114 bp, respectively, which were only one A base apart (Table 2). This sequence repetition is not unique. There are four repetitive sequences of 118 bp in length in *T. sejilaensis*, four 107 bp repeat sequences in *A. yunnanensis*, eight 113 bp repeats in *T. renzhiensis* [36], six 112 bp repeats in *T. gonggaensis* [39] and five 119 bp repeats in *T. pui* [37].

### 3.5. Phylogenetic Relationships and Taxonomic Relation

To confirm the evolutionary position of the host insects of *O. sinensis*, a phylogenetic tree of 11 species using published mitochondrial genomes (10 Hepialidae, 1 outgroup) and three genomes from the present study was constructed based on the concatenated nucleotides’ alignment of 13 PCGs or each PCG by the ML method. SD♂ × SD♀ was classified into a separate clade and GG♂ × GG♀ and SD♂ × GG♀ into another separate clade (Figure 5 and Appendix A). Thus, SD♂ × SD♀ and GG♂ × GG♀ were confirmed to be different species. SD♂ × SD♀ was considered to be an undescribed *Thitarodes* species according to the present database, and GG♂ × GG♀ was identified as *T. shambalenensis* by the *cox1* fragment. The genetical characteristics of SD♂ × GG♀ were close to those of GG♀, GG♂.

## 4. Discussion

*T. shambalaensis* (GG♂ × GG♀) identified by morphology (male genitalia) and genetic *cox1* phylogeny [43] and an undescribed *Thitarodes* species (SD♂ × SD♀) live in different locations in the Tibetan Plateau. In the present study, interspecific hybridization between these two distinct ghost moth species was demonstrated in the laboratory. The developmental performance of the studied *Thitarodes* populations was influenced by the hybridization, whereas the larval sensitivity to the fungal infection of the inbred populations was affected by the parental populations.

Hybridization between two related insect species is common in the laboratory or in the field; for instance, interspecific hybridization has been reported between *Helicoverpa armigera* and *Helicoverpa assulta* [57], *Nasutitermes corniger* and *Nasutitermes ephratae* [58], *Coptotermes formosanus* and *Coptotermes gestroi* [59] and between *Reticulitermes flaviceps* and *Reticulitermes chinensis* [43]. The red imported fire ant *Solenopsis invicta*, black imported fire ant *Solenopsis richteri* and their hybrid (*S*. *invicta* × *S*. *richteri*) are present in the field in Tennessee, USA [60]. The studied *T. shambalaensis* and *Thitarodes* sp. do not share a habitat in the Tibetan Plateau. It is speculated that reproductive individuals of the two ghost moth species may not have the chance to hybridize in nature because of the limited flying capacity of the adults. Surprisingly, these two species could mate, and the resulting hybrids produced a next generation. Whereas the inbred SD♂ × SD♀ laboratory population was very weak (I = 0.01), the SD♂ × GG♀ population exhibited a higher population trend index (I = 0.32), implying growth potential to a certain extent from GG♂ × GG♀, whose inbred population had a higher population trend index (I = 2.25). Hybridization may thus provide an effective method to create *Thitarodes*/*Hepialus* populations with increased growth potential for the improved artificial production of the insect hosts. 

Why these distinct *Thitarodes* species can be hybridized in the laboratory remains unknown. Species are defined to be groups of interbreeding natural populations that are reproductively isolated from other such groups [61]. The mechanism of pre-zygotic or post-zygotic reproductive isolation is considered to be involved in speciation [62]. Pre-zygotic reproductive isolation includes ecological and geographical habitat isolation, mating season or time difference, genitalia structure isolation, gamete isolation and mating or mating behavior isolation, whereas post-zygotic reproductive isolation includes survival limitation, infertility and depression of the hybrids [62]. Certainly, in this study, the hybridization of two *Thitarodes* species occurred in the laboratory, not in nature. The resulting hybrids also produced a next generation, indicating that the post-zygotic reproductive isolation may not prevent hybridization between two *Thitarodes* species, even in nature. Thus, the successful hybridization of these two species should depend on overcoming the pre-zygotic reproductive isolation, especially geographical habitat isolation and mating behavior isolation. Although reproductive isolation can evolve in a number of different ways, species-specific mate recognition by sex pheromones is believed to be a key element [63]. Similar recognition systems are a prerequisite for the interspecific interactions of closely related species in nature. However, in the laboratory, heterospecific partners can compulsively interact without the species-specific mate recognition. It appears that these two *Thitarodes* species can overcome the different genitalia structure and gamete isolation in the pre-zygotic phase and reproductive isolation in the post-zygotic phase in the laboratory. These results demonstrate the complexity of reproductive isolation and provide useful cues for further study in the speciation mechanism. 

The three complete mitochondrial genomes of GG♂ × GG♀, SD♂ × SD♀ and SD♂ × GG♀ differ not only in the size of the genome but also in the A + T-rich region with repeat sequences. So far, eight *Thitarodes*/*Hepialus* mitochondrial genomes are sequenced, including *T. renzhiensis*, *T. yunnanensis*, *T. pui*, *H. xiaojinensis*, *H. gonggaensis*, *T. sejilaensis*, an undescribed *Thitarodes* sp. and *T. damxungensis* [21]. Based on the phylogenetic tree constructed by 13 PCGs from the previously described genomes, GG♂ × GG♀ was identified as *T. shambalenensis*, and SD♂ × SD♀ was considered to be an undescribed *Thitarodes* species (Figure 5), given the reasonable threshold for inter-species variation (2.5% genetic distance) [43,64]. Interestingly, SD♂ × GG♀ was close to GG♂ × GG♀, according to the genetical similarities, which confirms the maternal inheritance of mitochondrial DNA. The sizes of eleven *Thitarodes*/*Hepialus* mitochondrial genomes, including the three genomes in the present study, are variable from 15,290 bp in *T. sejilaensis* [40] to 16,280 bp in *Thitarodes* sp. [41]. Likewise, reports that the mitochondrial genomes of the hybrids of bream fishes [65] or *Acipenser schrenckii* (♀) × *Huso dauricus* [66] are variable in gene sizes. Why the *Thitarodes* hybrid and the populations sharing the same mother have different mtDNAs in genome length, A + T content and the sizes of the A + T-rich region needs further study. 

The sizes of the A + T-rich region are 787 bp with two repetitive sequences of 114 bp, 554 bp without repetitive sequences and 673 bp without repetitive sequences in GG♂ × GG♀, SD♂ × SD♀ and SD♂ × GG♀, respectively. The A + T region of SD♂ × SD♀ contains non-repetitive sequences only, just like that in *T. damxungensis* [21]. It seems that the sizes of the A + T-rich region with repetitive sequences are quite different among the available Hepialidae mitogenomes, e.g., those in *T. damxungensis* (545 bp without repetitive sequences), *T. pui* (1030 bp with five 119 bp repeat units), *T. sejilaensis* (484 bp with four 118 bp repeat units), *T. yunnanensis* (1000 bp with four 107 bp repeat units), *T. xiaojinensis* (634 bp with four 118 bp repeat units), *T. gonggaensis* (1133 bp with six 112 bp repeat units), *T. renzhiensis* (1358 bp with eight 113 bp repeat units) and an undescribed *Thitarodes.* sp. (1472 bp with nine 112 bp repeat units) [21]. The relatively fast evolutionary rate in this A + T region appears to cause significant size variation [41].

## 5. Conclusions

In conclusion, the present study demonstrated that interspecific hybridization occurred under laboratory conditions between two allopatric and morphologically distinct ghost moth species *T. shambalaensis* and *Thitarodes* sp. Secondly, we found that the offspring produced by hybridization may display increased growth potential at least from one of the parent populations, which would greatly improve the cultivation of *Thitarodes* insects for the artificial production of Chinese cordyceps. Finally, our study demonstrated that the mitochondrial genome from the hybrid is different from those of its parents in several features (genome length, A + T content and the sizes of the A + T-rich region) and maternal inheritance.

## Figures and Tables

**Figure 1 insects-12-01046-f001:**
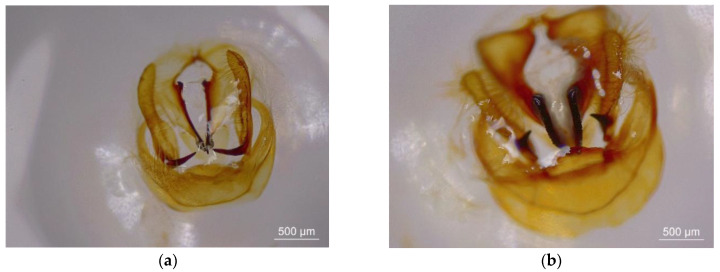
The valve patterns in male genitalia of *Thitarodes* sp. (SD♂ × SD♀) (**a**) and *Thitarodes shambalaensis* (GG♂ × GG♀) (**b**). Bars = 500 µm.

**Figure 2 insects-12-01046-f002:**
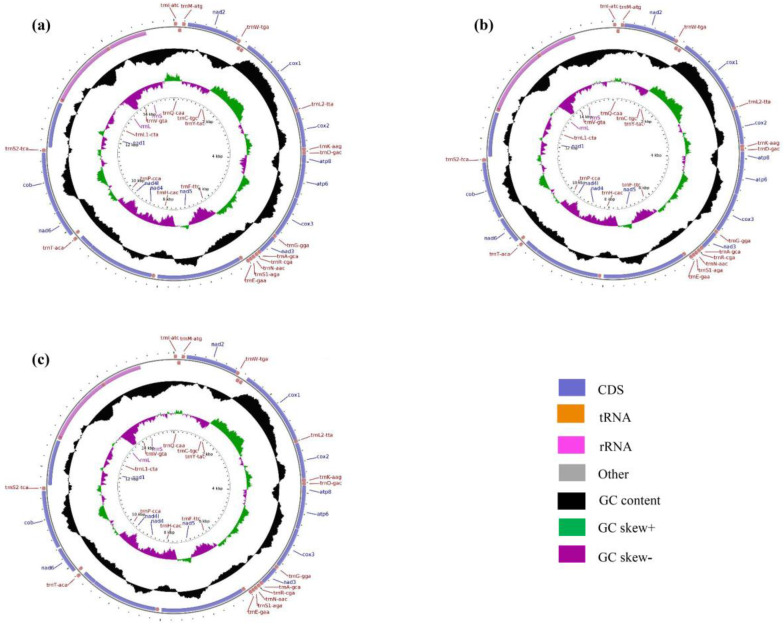
Circular map of the mitochondrial genomes of *Thitarodes* populations: SD♂ × SD♀ (**a**), GG♂ × GG♀ (**b**) and SD♂ × GG♀ (**c**).

**Figure 3 insects-12-01046-f003:**
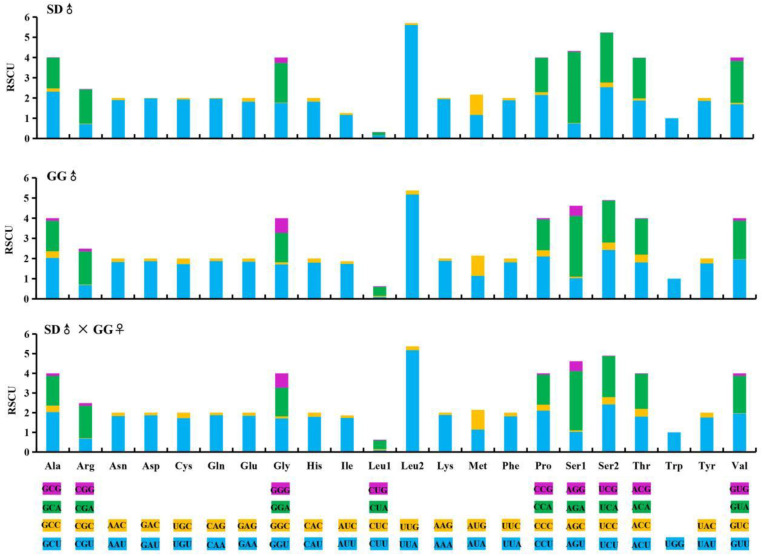
Codon usage numbers (N) and the relative synonymous codon usage (RSCU) among three *Thitarodes* mitochondrial genomes.

**Figure 4 insects-12-01046-f004:**
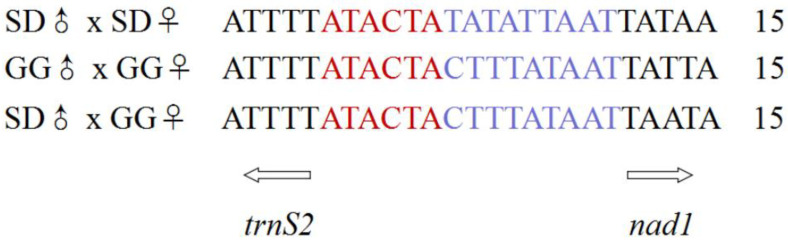
The highly conserved intergenic spacer located between the *trnS2* and *nad1*.

**Figure 5 insects-12-01046-f005:**
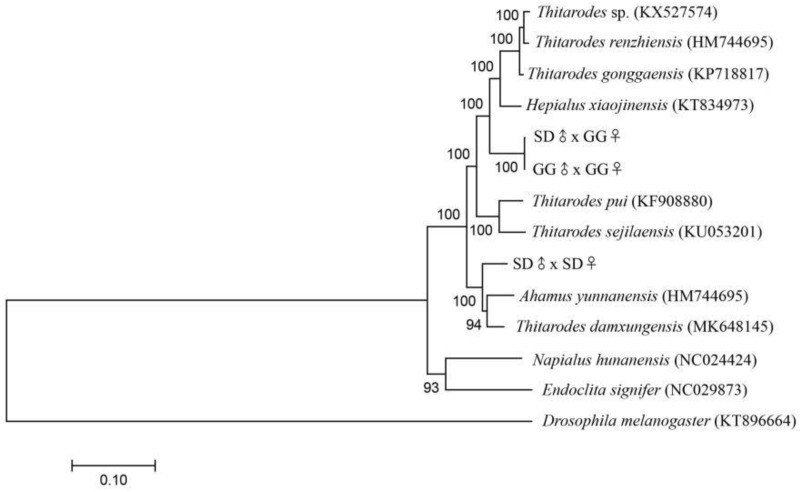
The phylogenetic relationship constructed by the amino acid sequence derived from 13 PCGs among 13 mitogenomes of Hepialidae insects and 1 outgroup.

**Table 1 insects-12-01046-t001:** Morphological and biological characters of *Thitarodes* sp. and *T.*
*shambalaensis*.

Strains		*Thitarodes* sp.	*T.* *shambalaensis*	*p*-Values
Pupae	Female fresh weight (g)	0.88 ± 0.04 a	0.67 ± 0.03 b	*p* = 0.014
	Male fresh weight (g)	0.59 ± 0.05 a	0.45 ± 0.03 a	*p* = 0.271
	Female body length (cm)	2.73 ± 0.21 a	2.50 ± 0.05 a	*p* = 0.335
	Male body length (cm)	2.46 ± 0.08 a	2.25 ± 0.04 a	*p* = 0.404
	Ratio of females and males	1.27 ± 0.16 a	1.04 ± 0.06 a	*p* = 0.254
	Female emergence rate (%)	38.38 ± 5.44 a	48.96 ± 2.75 a	*p* = 0.231
	Male emergence rate (%)	43.57 ± 6.79 a	57.58 ± 2.19 a	*p* = 0.188
	Color	Light to black yellow	Light to dark yellow	
Adults	Female fresh weight (g)	0.48 ± 0.02 a	0.42 ± 0.04 a	*p* = 0.223
	Male fresh weight (g)	0.20 ± 0.01 b	0.22 ± 0.01 a	*p* = 0.013
	Female body length (cm)	2.77 ± 0.03 a	2.77 ± 0.07 a	*p* = 1.000
	Male body length (cm)	2.13 ± 0.03 b	2.43 ± 0.03 a	*p* = 0.003
	Ratio of females and males	0.89 ± 0.06 a	0.87 ± 0.03 a	*p* = 0.800
	Female longevity (day)	6.0 ± 0.6 a	5.3 ± 0.3 a	*p* = 0.374
	Male longevity (day)	6.0 ± 0.6 a	5.3 ± 0.3 a	*p* = 0.374
	Oviposition period (day)	5.3 ± 1.2 a	4.3 ± 0.3 a	*p* = 0.621
	Mating time	Evening and night	All day and night	
	Females	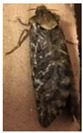	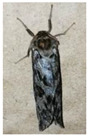	
	Males	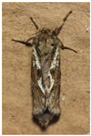	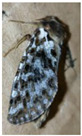	

Note: the rows with the same letters indicated no significant differences (*p* > 0.05).

**Table 2 insects-12-01046-t002:** Comparison of the complete mitochondrial genomes of three *Thitarodes* populations.

**Mitochondrial Genome**		**SD** **♂**					**GG** **♂**					**SD** **♂ × GG** **♀**				
Overall length (bp)		15,389					15,612					15,496				
A%		41.51					41.26					41.22				
T%		40.84					39.66					39.64				
C%		10.43					11.75					11.78				
G%		7.22					7.33					7.36				
A + T%		82.35					80.92					80.87				
AT skew = (A − T)/(A + T)		0.008					0.020					0.020				
GC skew = (G − C)/(G + C)		−0.182					−0.232					−0.231				
PCGs: length (bp)		11,073					11,067					11,067				
PCGs: A + T%		80.78					79.00					78.97				
tRNA: length (bp)		1474					1478					1477				
tRNA: A + T%		84.46					83.42					83.41				
rRNA: length (bp)		2132					2114					2114				
rRNA: A + T%		85.60					85.10					85.10				
A + T-rich length (bp)		554					787					673				
A + T-rich A + T%		91.70					89.83					90.64				
A + T-rich base constitution		No repeat sequence. 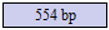					Two repeat base sequences. A = 113 bp, B = 114 bp, C = 114 bp 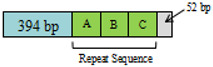					Base sequences. A = 113 bp, B = 114 bp 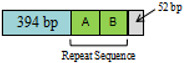				
Composition		13PCGs +22tRNAs + 2rRNAs + non-coding region														
intergenic arrangement		9 PCGs and 14 tRNAs were in the F chain, and 4 PCGs, 8 tRNAs and 2rRNAs were in the R chain. The arrangement of each gene in the three self-measured mitochondrial genomes was consistent.														
**Gene**	**Chain**	**Location**	**Spacer**	**Length**	**Initiation code**	**Termination code**	**Location**	**Spacer**	**Length**	**Initiation code**	**Termination code**	**Location**	**Spacer**	**Length**	**Initiation code**	**Termination code**
*trnI*	F	1–67	-	67			1–65	-	65			1–64	-	64		
*trnQ*	R	78–146	10	69			63–131	−3	69			62–130	−3	69		
*trnM*	F	158–227	11	70			151–220	19	70			148–217	17	70		
*nad2*	F	264–1247	36	984	ATA	TAA	257–1240	36	984	ATA	TAA	254–1237	36	984	ATA	TAA
*trnW*	F	1249–1314	1	66			1239–1304	−2	66			1236–1301	−2	66		
*trnC*	R	1307–1373	−8	67			1297–1367	−8	71			1294–1364	−8	71		
*trnY*	R	1379–1444	5	66			1374–1440	6	67			1371–1437	6	67		
*cox1*	F	1447–2977	2	1531	CGA	T	1443–2973	2	1531	CGA	T	1440–2970	2	1531	CGA	T
*trnL2*	F	2978–3046	0	69			2974–3042	0	69			2971–3039	0	69		
*cox2*	F	3049–3730	2	682	ATG	T	3045–3726	2	682	ATG	T	3042–3723	2	682	ATG	T
*trnK*	F	3731–3801	0	71			3727–3797	0	71			3724–3794	0	71		
*trnD*	F	3801–3866	−1	66			3797–3861	−1	65			3794–3858	−1	65		
*atp8*	F	3867–4031	0	165	ATA	TAA	3862–4023	0	162	ATA	TAA	3859–4020	0	162	ATA	TAA
*atp6*	F	4028–4702	−4	675	ATA	TAA	4020–4694	−4	675	ATA	TAA	4017–4691	−4	675	ATA	TAA
*cox3*	F	4705–5490	2	786	ATA	TAA	4697–5482	2	786	ATA	TAA	4694–5479	2	786	ATA	TAA
*trnG*	F	5493–5558	2	66			5485–5550	2	66			5482–5547	2	66		
*nad3*	F	5559–5912	0	354	ATT	TAG	5551–5904	0	354	ATT	TAG	5548–5901	0	354	ATT	TAG
*trnA*	F	5911–5978	−2	68			5903–5971	−2	69			5900–5968	−2	69		
*trnR*	F	5982–6047	3	66			5975–6040	3	66			5972–6037	3	66		
*trnN*	F	6055–6120	7	66			6045–6110	4	66			6042–6107	4	66		
*trnS1*	F	6121–6181	0	61			6111–6170	0	60			6108–6167	0	60		
*trnE*	F	6182–6247	0	66			6171–6235	0	65			6168–6232	0	65		
*trnF*	R	6258–6324	10	67			6238–6304	2	67			6235–6301	2	67		
*nad5*	R	6325–8020	0	1696	ATA	T	6305–7997	0	1693	ATA	T	6302–7994	0	1693	ATA	T
*trnH*	R	8063–8129	42	67			8043–8109	45	67			8040–8106	45	67		
*nad4*	R	8131–9471	1	1341	ATG	TAA	8111–9451	1	1341	ATG	TAA	8108–9448	1	1341	ATG	TAA
*nad4L*	R	9471–9746	−1	276	ATA	TAA	9451–9726	−1	276	ATA	TAA	9448–9723	−1	276	ATA	TAA
*trnT*	F	9767–9832	20	66			9747–9812	20	66			9744–9809	20	66		
*trnP*	R	9833–9897	0	65			9813–9876	0	64			9810–9873	0	64		
*nad6*	F	9900–10,424	2	525	ATA	TAA	9879–10,403	2	525	ATA	TAA	9876–10,400	2	525	ATA	TAA
*cytb*	F	10,424–11,569	−1	1146	ATG	TAA	10,403–11,548	−1	1146	ATG	TAA	10,400–11,545	−1	1146	ATG	TAA
*trnS2*	F	11,575–11,645	5	71			11,557–11,629	8	73			11,554–11,626	8	73		
*nad1*	R	11,661–12,572	15	912	ATA	TAA	11,645–12,556	15	912	ATA	TAA	11,642–12,553	15	912	ATA	TAA
*trnL1*	R	12,594–12,662	21	69			12,578–12,648	21	71			12,575–12,645	21	71		
*rrnL*	R	12,638–13,992	−25	1355			12,649–13,984	0	1336			12,646–13,981	0	1336		
*trnV*	R	13,993–14,057	0	65			13,984–14,048	−1	65			13,982–14,046	−1	65		
*rrnS*	R	14,059–14,835	1	777			14,048–14,825	−1	778			14,046–14,823	−1	778		
*A + T-rich*		14,836–15,389		554			14,826–15,612		787			14,824–15,496		673		

Note: F and R refer to the majority and minority strands, respectively. Positive values of intergenic regions indicate gap nucleotides; a negative value indicates overlapped nucleotides. Position numbers refer to positions on the majority strand.

## Data Availability

Exclude this statement.

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
