# Peer review of "Interspecific Hybridization and Complete Mitochondrial Genome Analysis of Two Ghost Moth Species"

_insects, 2021, doi:10.3390/insects12111046_

Round 1

Reviewer 1 Report

Following my previous remarks, the Authors have addressed my critical points. I believe the manuscript is now ready to be published.

Author Response

Thank you for your suggestions and comments.

Reviewer 2 Report

In this resubmitted manuscript, the authors have improved some parts of the previously submitted manuscript. However, I still have three major concerns (see below) about the results and structure of the manuscript. Thus I feel that the results in this resubmitted manuscript seem to lack general interest to the field of entomology and the interest as scientific topic in results shown in this manuscript is narrow and limited. Hence this manuscript seems to be not suitable for Insects. I suggest the authors better reorganize their results and submit to a more specific journal.

Major comments:

(1) The authors have improved the introduction, discussion, and conclusion. However, I still could not find relevance of three results, Morphological and biological characters of Thitarodes sp. and T. shambalaensis (section 3.1. and 3.2.), Larval Infection of Inbred Populations by O. sinensis Isolates (section 3.3.) and Mitochondrial Genome Analysis (section 3.4.). I understand that the results in each section would represent some novelty in the limited field of entomology. However, the authors did not address how the differences in mitochondrial genome sequences between inbreed and hybrid populations affect their morphological and biological traits or the sensitivity of larvae to fungal infection. If the authors wish to present the above three results in one manuscript, to clarify this issue should be very important.

(2) L18-L20, L37-L39 and L536-L537: The authors claimed that increased hybrid vigor as compared with the parental Thitarodes sp. (SD) population and that the offspring produced by hybridization may display heterosis. However, according to Scheme 2 (Figure S2), it appears that there are no significant differences in larval survival rate between inbred (SD) and hybrid populations. The values of the population trend index shown in L313-L315 seems to be different in the inbred and hybrid populations. If the vigor of the hybrid population is better than that of the SD and GG populations, the authors can claim in L479-L481, “Hybridization may thus provide an effective method to create Thitarodes/Hepialus populations with hybrid vigor for the improved artificial production of the insect hosts” and conclude, “heterosis, which would greatly improve the cultivation of Thitarodes insects for the artificial production of Chinese cordyceps” (L536-L538). However, Scheme 2 (Figure S2) and the population trend index values showed that one of the parental populations (GG) was better than the offspring produced by hybridization. Hence these results shown in this manuscript do not support the author’s claim shown in L479-L481 and L536-L538.

(3) L322-L335 and Table S2 and S3: The authors showed the results of larval infection only in inbred populations. How is the sensitivity of larvae to fungal infection of the hybrid population? In this manuscript, the authors introduced that “the described insect host species/or strains of O. sinensis have never been confirmed by a hybridization technique” (L78-L79) and claimed that “The biological (developmental performance and larval sensitivity to the fungal infection) characteristics of the resulting inbred and hybrid populations were influenced by the hybridization” (L463-L465). However, the authors did not address the sensitivity of larvae to fungal infection of the hybrid population. Why did not the authors address this point of the hybrid population?

Minor comments:

L555, L559, L565, L568, L603: change “Scheme1-5” to “Figure S1-S5”

L253-L254, population trend index (I): I’m not sure whether the description and formula are correct or not. According to the description and the formula (I=PI/PII), if the values (I) are low under 1, the numbers of pupa in the next generation should be greater than that in the parental generations. However, the authors described “SD population was very weak (I=0.01),” in L476-L479. If the description and formula are correct, this means (I=0.01) that the authors had 100 times more pupae in the next generation than in the parental generations. How can the authors claim that this is very weak?  

Reviewer 3 Report

This is a follow-up to my previous review. I really liked the attention that the authors put into addressing my previous concerns. They were quite thoughtful and thorough and the manuscript is much more cohesive and complete now (in my opinion). I have no problems with this going forward, as is, to publication. 

Editorial note: Line 391, change "night" to "eight". 

Thank you for lines 516-520! That was what was occurring to me--why would the hybrid be different at all? Could this be sequencing error? Could it be simple intraspecific variation (e.g. if the SDmale X GCfemale was not EXACTLY the offspring of the GCmale X GCfemale, then these differences could simply be due to intraspecific variation carrying through to different parentage). 

Author Response

Point 1: Line 391, change "night" to "eight".

Response 1: Done. “night” is changed to “nine” (Line 402).

Round 2

Reviewer 2 Report

In this revised manuscript, the authors have improved some parts of the previously resubmitted manuscript. However, I still believe that the results in this revised manuscript seem to lack general interest to the field of entomology. As I mentioned in previous review comments, the authors did not address how the differences in mitochondrial genome sequences between inbreed and hybrid populations affect their morphological and biological traits or the sensitivity of larvae to fungal infection. From an entomological or scientific point of view, I believe that clarification of the above issues should be important in this manuscript. Otherwise, the interest in the results of each section as a scientific topic is narrow and limited. Hence this manuscript seems to be not suitable for Insects. I suggest that the authors better reorganize their results and submit to a more specific journal.

This manuscript is a resubmission of an earlier submission. The following is a list of the peer review reports and author responses from that submission.

Round 1

Reviewer 1 Report

In some ways I really liked this paper. The motivation behind the study--to develop a system in which Ophiocordyceps could more easily and predictably be cultivated--was well developed; and the idea to investigate the possibility that hybrids between hosts could serve this was intriguing. The paper was well-written and was overall very well-presented. 

I have two fundamental critiques of the paper, and then some minor suggestions that the authors can readily address. The fundamental critiques do not attempt to negate the scientific validity of the papers; instead, they point to some disconnectedness throughout the manuscript.

My first critique is in the connection between what are two VERY different foci of the manuscript. There are really two papers here: (1) the outcomes of hybridization and the interplay between hybrids and Ophiocordyceps infection; and (2) the comparative mitochondrial genomics of the two parental species. There really was no integration between the two here (except to show that the mitochondrial genome is maternally inherited, which is a rather mundane result). The introduction of why the mitochondrial genomics was done is almost non-existent and should be improved. Mechanistically, both parts of this were done quite well, mind you; but they simply do not integrate. I would recommend either that the authors treat these as two separate papers OR that the authors are quite explicit that their manuscript has two VERY different goals. 

My second critique is a lack of integration between the introduction (e.g. might hybrids be a potential source of culturing Ophiocordyceps) and the conclusions (which barely discuss this). I would like to see a more full discussion of what their results have to say about this. It might even be something like "These results are very preliminary on whether hybrids will be useful in developing cultivation of Ophiocordyceps, because x, y, and z still need to be done. However, we have shown here that a, b, and c indicate this is a promising avenue of further research (or not)". 

With all of this said, the two very different parts of the paper are very well done. The description of the hybridization experiments and results is clear and the outcomes are well presented. The description of the comparative genomics is overall outstanding. 

To improve the paper, I would recommend two minor modifications. First, I would recommend that the authors clarify the statistical results in Table 1. The authors could simply include a statement in the caption that results considered significantly different are indicated by different letters. Having said that, I put very little stock in the magical "p < 0.05 is significant". I would prefer to interpret the results myself by actually seeing the exact p-values (which are readily provided by SPSS). A p = 0.00001 is a very different result from p = 0.049; and I would also like to be able to judge whether or not a result considered "not significant" is perhaps trending (e.g. p = 0.07) and might simply be a Type II error due to lack of power or high levels of variation. I would recommend adding a fourth column to Table 1 with that value. The authors can then mention that for the purpose of their conclusions/discussion they will consider anything p < 0.05 to be significantly different. 

The second issue I have is with the phylogenetic analysis. I realize that this is not the main focus of the paper, but it is poor at best. Mitochondrial genomes are rather complex entities. To analyze the entire genome using the "Mega default settings" indicates a real lack of rigor. First off, the authors should at least include what that default setting is (Jukes-Cantor). Given the high variation in base-pair bias across the genome (which they show), and likely even more bias when protein-coding genes are separated by codon, the Jukes-Cantor default setting is almost certainly a horrible choice for phylogeny reconstruction. While it is possible that the choice will not dramatically impact the topology of the phylogeny, it is also possible that it will. Mind you, if the only point of the analysis is to show that the mitochondrial genome is inherited maternally this is largely moot and could probably be done by a simple identity analysis, and not a phylogenetic one. 

Reviewer 2 Report

In this manuscript, the authors present morphological and biological characters of ghost moths, Thitarodes sp., T. shambalaensis and their hybrids, and their complete mitochondrial genome sequencing. Using some genes of the complete mitochondrial genomes, the authors reconstructed molecular phylogenetic tree in 13 Hepialidae insects and clarified the phylogenetic position of species studied.

It seems the manuscript are not well organized, because I could not find any relevance of three results, Morphological and biological characters of Thitarodes sp. and T. shambalaensis (section 3.1. and 3.2.), Larval Infection of Inbred Populations by O. sinensis Isolates (section 3.3.) and Mitochondrial Genome Analysis (section 3.4.). Why did the authors show these three results in a manuscript together? T. shambalaensis seems to be already identified by molecular marker of a part of mtDNA. Why are the mitochondrial genomes from the hybrids important? The authors conclude the maternal inheritance of mitochondrial DNA but this should be well known in most of organisms. It was difficult for me to understand the motivation of this manuscript.

I understand the research of interspecific hybridization should be interesting from point of view of ecology or evolutionary biology or genetics etc., mitochondrial sequences should be useful and important for clarifying phylogenetic position of species studied and identifying species etc. However I could not find any novelty from scientific point of view in this manuscript. Results shown in this manuscript seem to lack general interest for entomology field to study.

I think the interest as scientific topic in results shown in this manuscript is narrow and limited, and the presentation of this manuscript are not well done. Hence this manuscript seems to be not suitable for Insects. I suggest authors better reorganize their results and submit to more specific journal. The following is my specific comments.

Page 1: Simple Summary seems to be totally same to Abstract. Omit Simple Summary.

L290, L312, L324 Table S1-S3: I could not find them in this manuscript file, so I could not evaluate them.

Reviewer 3 Report

The interspecific hybridization and the description of the three complete mitochondrial genome of two ghost moth species, important host species for the Chinese cordyceps, are investigated in this manuscript.

The topic is interesting and has repercussions in the economic sphere, although locally. The manuscript is noteworthy although specific points are not clearly described and some methods not described at all.

First of all, I don’t understand why the hybrid and specimen that share the same mother female species have mtDNAs that are so different and vary for the length and other molecular details. The topic is not discussed and needs further in-depth study.

This manuscript doesn’t address the topic and some points are more incomprehensible due to the lack of a clear description on how the sequences were generated. Apart from the normal variability that one might expect from the comparison of two mitochondrial genomes from two female specimens belonging to the same species (intraspecific variability), if maternal inheritance of mitochondria is a rule for Eukarya, it is not justifiable that there are so many differences between hybrid and parent species, that share the same mother species. Differences in terms of length of genes and spacers, but also in nucleotide composition and more. My guess is that the sequencing method and subsequent assembly analysis of the reads were not so accurate, and that generated significant nucleotide differences between genomes from mothers belonging to the same species, due to errors. This topic needs to be clarified / discussed based on the analysis conducted to obtain the mtDNAs. Unless the authors should provide a plausible justification that escapes my mind at this time.

In addition, the choice of taxa (taxa selection) for phylogenetic analysis is not provided, as well as methods of models of nucleotide substitution applied for ML analyses.

Other minor remarks are listed within the pdf file attached to this review.

In conclusion, Authors should address the raised topics before the manuscript could be evaluated again for publication.
